# Impact of Oncology Pharmacists on the Knowledge, Attitude, and Practices of Clinicians to Enhance Patient Engagement of Self-Administered Oral Oncolytics

**DOI:** 10.3390/pharmacy9030130

**Published:** 2021-07-23

**Authors:** Shannon Palmer, Ashley Chen, Taylor Dennison, Cameron Czech, Jessica Auten, Kaitlyn Buhlinger, Benyam Muluneh

**Affiliations:** 1Department of Pharmacy, University of North Carolina Medical Center, Chapel Hill, NC 27514, USA; palmersh@ohsu.edu (S.P.); awchen@seattlecca.org (A.C.); taylordennison8@gmail.com (T.D.); camczech02@gmail.com (C.C.); jessica.auten@unchealth.unc.edu (J.A.); kaitlyn.buhlinger@unchealth.unc.edu (K.B.); 2UNC Eshelman School of Pharmacy, University of North Carolina, Chapel Hill, NC 27514, USA

**Keywords:** patient activation, patient engagement, behavioral health change, self-administered therapy, oral oncolytic

## Abstract

Oncology clinical pharmacists are uniquely positioned to make interventions to impact the knowledge, attitudes, and practices of clinicians as well as patient activation and engagement. To accomplish this goal, pharmacists can target health system-related, provider-related, and patient-related factors to enhance patient-centered care and drive behavioral health changes. Interventions that pharmacists must tackle include educating team members and patients on the medication acquisition process, communicating urgency of treatment, optimizing workflows, facilitating guideline recommendations, preventing, and managing treatment toxicities, and promoting patient self-advocacy through education and shared decision-making. As crucial members of the healthcare team, oncology pharmacists can simplify highly complex treatment regimens to facilitate and optimize patients’ ownership of their care. This narrative review will focus on the example of venetoclax treatment in acute myeloid leukemia to demonstrate the impact that pharmacists provide that leads to behavioral change of patients and clinicians.

## 1. Introduction

With the rise of highly complex self-administered therapies in cancer care, it is critical to have a structured approach for promoting patient activation and engagement. The concepts of patient activation and engagement provide a framework to help increase adherence and improve outcomes for patients. Patient activation is a term used to denote the ability, knowledge, and confidence of the patient to be an active participant in their health care. Patient engagement, on the other hand, is a broader term that encompasses activation, interventions to increase activation, and the eventual behavioral change that may lead to improved health outcomes [1,2]. With the movement toward value-based care, it is crucial for the medical team to help remove barriers preventing patient engagement. One strategy that demonstrates interventional targets is the World Health Organization (WHO)’s adherence model. The WHO adherence barriers that can be mitigated through behavioral interventions include health system-related, provider-related, and patient-related factors [3]. By addressing these three domains, oncology pharmacists can optimize workflows to enhance patient-centered care and drive behavioral health changes. To address the health system domain, pharmacists can communicate urgency to payors and specialty pharmacies to ensure rapid initiation and approval of medications. Pharmacists influence the provider-related domain through facilitating guideline recommendations and navigating toxicities by shifting the knowledge, practice, and attitude of the healthcare team. Finally, pharmacists impact the patient-related domain by ensuring patients understand the complexities of their regimens and achieve patient activation. We illustrate the implementation of the WHO framework by oncology pharmacists using acute myeloid leukemia (AML) patients on oral oncolytics as an example.

The incorporation of oral oncolytics and subsequent shift in cancer treatment to the outpatient setting has created a need to shift the behavior of the medical team and the patient to accomplish therapeutic goals. One of the primary barriers to achieving success is potentially low adherence to oral agents. A review conducted to examine interventions to enhance adherence to oral antineoplastics demonstrated that out of 56 articles fewer than one-half (44.7%) of the trials with comparator arms were able to demonstrate significant improvement in adherence with their chosen intervention. However, pharmacist-directed programs especially ones that had integrated monitoring or close provider follow-up, did appear to have efficacy [4]. Therefore, applying oncology pharmacists’ skillsets with oral oncolytic use may be an avenue to improved clinical results. This current review aims to demonstrate interventions that the oncology pharmacist may initiate for oral oncolytic programs specifically.

One oral oncolytic used in AML requiring significant oncology pharmacist oversight and intervention is venetoclax. Venetoclax is a B-Cell Lymphoma 2 (BCL-2) inhibitor that is used for the treatment of newly diagnosed AML in patients who are ineligible for intensive chemotherapy [5,6,7]. Many barriers exist to initiating venetoclax in the outpatient setting including complexity of dosing schedule, drug interactions, medication access, and toxicity prevention and management. Using venetoclax-based treatment regimens as an example, we highlight the role of the pharmacist as a catalyst affecting health behavior through their interactions with patients, their caregivers, and other health care providers. Pharmacists are well positioned to impact the three WHO domains-health system-related, provider-related, and patient-related barriers—to help promote patient engagement toward more positive clinical outcomes.

## 2. Health System-Related Domain

The first step to achieving patient engagement with venetoclax regimens includes helping the patient navigate medication access to prevent financial toxicity, the negative impact medical expenses can have on a patient, and to rapidly initiate therapy. Many socioeconomic factors are considered when coordinating care for these patients and pharmacists are well positioned to facilitate throughout this process [8]. This often requires pharmacists to educate relevant stakeholders regarding the importance of urgent drug acquisition and initiation of therapy given the aggressive biology of AML.

Shepherding Medication Access and Ensuring Rapid Initiation of Therapy

Healthcare providers who prescribe and manage specialty medications often struggle to navigate the complex process of medication authorization and copay assistance [9,10,11]. Successful acquisition of affordable medication is dependent on a variety of patient and health system-related factors (Figure 1). Given the high cost of venetoclax and limited distribution networks, insurance providers usually require completion of a prior authorization. Once prior authorization is approved, the cost associated with venetoclax therapy may still be financially prohibitive for patients. Pharmacists must actively work and communicate with medication access teams to identify potential copay assistance resources and help ease the financial toxicity associated with therapy. These resources include copay assistance cards for commercially insured patients, national organizational funds (also called grants) often used for Medicare patients, or manufacturer sponsored patient assistance programs frequently used for uninsured patients. Additionally, pharmacists are helpful in clarifying this complex process for patients and communicating the time-sensitive nature of AML therapy to payors to allow for prompt approval and initiation. For patients, this may be their first-time interfacing with a specialty pharmacy. Because of the complexity of the medication access process, the patient serving as his/her own advocate yields the best success. Pharmacists can ask patients questions to ensure they understand their role in this process, such as access to a fax machine or scanner if more financial information is needed, access to secure electronic messaging system for easy communication, and awareness of coordination for initial fill and subsequent fills to be delivered to their home. Since many patients receiving this regimen are older adults, methods other than electronic forms of communication should be considered. To obtain access efficiently, the pharmacist oversees the logistics while coaching patients to understand and remain fully engaged in the process. In an ideal state, medication access teams investigate access and copay assistance, thus removing that barrier entirely for patients. However, for institutions without dedicated medication assistance teams, pharmacists are often critical in facilitating proper communication between all stakeholders to ensure prompt access to therapy. Pharmacists can help create a resource with a systematic approach that can help other team members understand the steps to medication access, to help with both patient education and more timely acquisition for the patient.

Oncology pharmacists also play an active role coordinating key elements of care with members of the multidisciplinary team to ensure additional barriers are being considered and addressed. Given the extensive monitoring, transfusion requirements, and frequent follow-up visits needed, additional socioeconomic factors should be considered. These include addressing reliable access to transportation, lodging, emotional or psychiatric support, and adequate caregiver support prior to initiation of therapy. Due to the complexities associated with AML management, patients are often initiated on therapy at an academic health center with the possibility of co-management with a local oncologist after initiation. Coordinating care for AML patients between treatment sites requires effective communication with outside providers; oncology pharmacists’ aid in this transition of care process to ensure that therapy has been appropriately and safely continued. Considering the nature of AML, delays in the coordination of all aspects of care can lead to complications associated with the underlying disease (i.e., infection, transfusion dependency, hyperleukocytosis) which can increase the risk of hospitalizations and elevate direct and indirect costs to the patient and the entire health system. Although coordinating care is not a traditional role of the pharmacist, it does become an important piece to consider in the effective management of acute leukemia patients in the outpatient setting. Without the initial efforts by pharmacists to help patients gain confidence in their ability to afford and access their medications, full patient engagement would be more difficult throughout the remainder of their care.

## 3. Provider-Related Domain

Another crucial barrier that can prevent patient engagement is the vast amount of information that patients are expected to retain during their visits with their providers. Providers cover a large range of information including diagnosis, prognosis, and discussion of treatment options. As drug regimens are often complex, pharmacists can help providers ensure patients are safely and efficiently started on therapy with appropriate supportive care by facilitating applicable guideline adherence. Additionally, the pharmacist often plays a key role in navigating therapy toxicities and empowering the patient to help prevent and manage side effects of therapy. With venetoclax specifically, some examples of pharmacist intervention include consideration of antifungal medications, prevention of tumor lysis syndrome (TLS), management of myelosuppression, and dose adjustments for drug interactions.

### 3.1. Facilitating Guideline Recommendations

The expanding body of research often makes it challenging for providers to be aware of all details and differences in drug information from various guidelines and emerging literature [12]. This is especially true for providers not practicing in an academic medical center with a dedicated leukemia program. There are often multiple dosing regimens (package inserts, clinical trial protocols), several guidelines (National Cancer Comprehensive Network, Infectious Diseases Society of America), and new data from real-world practice that must be consolidated for use of therapy in any given patient. To close this gap, pharmacists can consolidate, analyze, and synthesize multiple guideline recommendations to help establish institutional standard approaches. Pharmacists can subsequently educate the medical team and other stakeholders to facilitate proper adoption and implementation of these guidelines. This document can be a “living” document, in which feedback is solicited from the medical team to modify guidelines with emerging data and consideration of the local context. Furthermore, retrospectively evaluated data can justify appropriate resources to optimize existing workflows.

One example of facilitating guideline recommendations related to venetoclax is pharmacist implementation of national guidelines for antimicrobial prophylaxis. Current guidelines recommend antibacterial prophylaxis for patients who are at high risk of infection, particularly patients who are expected to have profound and protracted neutropenia [13,14]. Prolonged episodes of neutropenia place patients at risk of severe infectious complications including invasive fungal infections. Due to this increased risk, prophylactic antifungal regimens such as triazole antifungals are routinely used in patients who are starting venetoclax-based regimens due to the high associated rate of grade 4 neutropenia [13,14]. The specific choice of antifungal agent used for prophylaxis varies between institutions and is often controversial. It is helpful for pharmacists to use trial data and institutional antibiograms to guide therapy selection. Efficacy, tolerability, venetoclax dose reduction strategies, education, and importantly medication access and cost of these agents should be discussed with the clinical care team by the pharmacist.

Pharmacists can also implement protocols to increase the safety of initiation of therapies such as venetoclax, such as a protocol for prevention of tumor lysis syndrome (TLS). Due to the rapid apoptotic effect and cellular responses seen with BCL-2 inhibition, TLS is a significant and potentially life-threatening complication associated with venetoclax-based regimens [15]. The package insert, as well as trial data, provide recommendations surrounding TLS mitigation, and pharmacists can help to standardize the application of such recommendations. Dose escalation strategies, close clinical and laboratory monitoring, and implementation of risk management protocols such as hydration and prophylactic regimens with urate-lowering therapies reduce the incidence of TLS in patients treated with venetoclax [6,16]. Initiating venetoclax using a dose escalation daily “ramp-up” strategy over the first three to four days to the target dose reduces the risk of TLS [17]. Laboratory evaluation should occur promptly by the provider and/or pharmacist during the venetoclax dose ramp-up period. There are several accepted definitions for TLS and pharmacists can help ensure consistency regarding thresholds for intervention by including instructions within therapy plans [18]. Additionally, with the initiation of venetoclax, pharmacists should educate providers on the vigilant monitoring and preventative measures they must relate to their patients to minimize the risk of TLS. Assessing patients’ oral hydration and encouraging them to drink the recommended daily 6–8 glasses of water while on this regimen is a critical component of initial patient education [15,17]. Beyond patient education, the pharmacist plays a central role in coordination with the team to ensure infusion orders include hydration and prescriptions are sent for urate-lowering therapies. By standardizing processes through protocols, pharmacists can ensure increased safety and less variability between patients initiated on therapy.

### 3.2. Navigating Therapy Toxicities

In addition to guideline and protocol development, pharmacists can assist with early recognition of potential drug interactions, therefore minimizing toxicity and harm to the patient. Venetoclax metabolism is primarily through the CYP3A4 system and venetoclax is a substrate of P-glycoprotein (P-gp), therefore, dose interactions occur when given concomitantly with CYP3A4 and P-gp inhibitors and inducers. Drugs that inhibit the metabolism of venetoclax can lead to supratherapeutic concentrations and potentially an increased risk of complications and adverse events. Dosing of venetoclax, further described in Table 1, is dependent on the strength of CYP3A4 inhibition. Venetoclax patients should also avoid ingestion of grapefruit products, as they inhibit CYP3A4 [19]. Due to the many potential drug interactions with venetoclax therapy, pharmacists can provide vital knowledge regarding the prevention and management of drug-drug interactions in these patients. Pharmacists can provide extensive education regarding these interactions to the medical team and emphasize the importance of reviewing any new medications in the context of the relevant drug-drug interactions.

Another challenge in navigating venetoclax therapy is myelosuppression, which can cause fatal complications [6,19]. Despite the potential severity of this side effect and the resulting necessity of treatment adjustments, specific and universally accepted recommendations regarding management are not available [6,19,21,22]. Furthermore, developing strategies and consensus is difficult as these approaches are typically institution-specific and dependent on providers’ preferences, as well as a patient’s place in therapy and treatment goals. The oncology pharmacist should assist in the development and maintenance of institutional guidelines to help standardize approaches to myelosuppression management that includes guidance related to granulocyte colony stimulating factor (GCSF) use, reducing the dose/durations of chemotherapy, and when disease assessment by bone marrow biopsy is indicated to guide treatment decisions. One strategy for myelosuppression management is to integrate breaks between cycles for patients with cytopenias [6,21,22]. Pharmacists can communicate with providers to employ strategies to increase patient activation by helping patients understand their therapy schedule and preventing patients from accidentally continuing venetoclax during therapy breaks. Several techniques can accomplish this, including drug calendars, patient reminders through the electronic health system, and written instructions provided at clinic visits. Although the venetoclax dose should not be reduced to manage myelosuppression, the venetoclax duration may be shortened (e.g., from 28 days to 21 days per cycle) in subsequent cycles or the hypomethylating agent (azacitidine, decitabine) dose intensity may be reduced if a patient has experienced prolonged hematologic recovery [6,19,21]. Given the potential for frequent schedule modifications, the pharmacist should review these adjustments with patients to ensure understanding and confirm prescriptions and orders reflect these changes. Additionally, management of myelosuppression with GCSF is an option; however, there is lack of consensus regarding the safest time to administer these agents [6,21]. If providers prescribe GCSF, pharmacists should confirm it is accessible and affordable for patients, as well as educate them on the proper storage and administration of GCSF. As GCSF administration is time-sensitive, pharmacists must clarify the schedule with patients and confirm their understanding.

## 4. Patient-Related Domain

The third domain that pharmacists can target to cultivate patient engagement is patient-related barriers, by instituting models such as shared decision-making. Shared decision-making employs choice, option, and decision talk to increase patient activation [23]. Decision talk refers to considering preferences of the patient and deciding what works best for them. Providing high quality information enables patients to act independently when provided with sufficient evidence about key issues. Pharmacists can provide this critical information and therefore augment patient engagement. Pharmacists empower patients to discuss formulation concerns, understand complicated treatment regimens, help monitor and manage side effects, and use strategies to increase medication adherence. By developing knowledge to overcome these barriers and engaging patients in shared decision-making, pharmacists enable patients to improve the likelihood that they will derive the greatest benefit from their oral chemotherapy.

### 4.1. Breaking Down Treatment Complexities

Medication formulations, especially the currently available venetoclax formulations, can lead to increased treatment complexity and decreased adherence. Venetoclax is available as a tablet and it should be swallowed completely without being chewed, crushed, or broken [19]. Venetoclax is available in 10 mg, 50 mg, and 100 mg tablets; however, only the 100 mg tablets are commercially available in bottles [19]. The smaller tablet sizes are packaged in bulky weeklong blister packs labeled and designed for chronic lymphocytic leukemia/small lymphocytic leukemia venetoclax regimens, making these packs impractical and unsafe to dispense to AML patients. Thus, for patients taking venetoclax for AML, the only practical dosage form available are the 100 mg tablets. However, when clinicians look to the package insert and phase 3 data for guidance in dose reductions in the setting of CYP3A4 inhibition, 70 and 50 mg doses are recommended when venetoclax is used in combination with posaconazole [6,19]. As this is not feasible in a real-world clinical setting for the reasons described above, patients are commonly prescribed the venetoclax 100 mg dosage form given the difficulty obtaining the smaller tablet sizes used in clinical trials [21,22]. A pharmacist can be helpful in describing the formulation challenges to their clinical team and provide standard guidance as to their institutional approach to venetoclax dosing. Additionally, pharmacists can engage patients in reasons for changing venetoclax dose, to ensure they understand no reduction in efficacy will occur when reducing venetoclax due to drug interactions.

Beyond dealing with formulation concerns, pharmacists’ roles and responsibilities become even more vital as anticancer therapies are shifting from intravenous infusions administered in the hospital or clinics to orally administered medications given by informal caregivers or the patients themselves [24]. Oral anticancer therapies empower patients to take responsibility for their own care and can improve patients’ quality of life but require patients to have an augmented understanding of their own treatment course to participate in decision talk [23]. Pharmacists can provide education to ensure this increased patient responsibility does not cause insurmountable adherence challenges and negative treatment outcomes [24]. If patients do not become fully activated and engaged, it increases the risk of non-adherence and subsequent consequences, including increased healthcare use (emergency room visits, hospitalizations), tumor resistance, increased patient and caregiver burden, disease progression, and even death [24].

Despite the critical importance of taking oral oncolytics such as venetoclax as prescribed, adherence to oral anticancer therapies is lower than adherence to intravenous anticancer therapies and ranges from 20–100% [24]. The reasons for patient non-adherence are multifactorial, including complex treatment regimens in addition to patients’ other prescribed medications, side effects from oral anticancer therapies, inconvenient clinic follow-up, abandoned prescription refills, inadequate supervision, suboptimal patient-clinician communication, complex socioeconomic factors, cost, misunderstanding of prescribed therapy, pill burden, pill aversion, and difficult packaging [24,25,26]. Pharmacists are positioned to improve adherence of complex regimens containing oral chemotherapy [27]. A key step in this process is education of the new regimen at the initial pharmacist-patient meeting prior to starting therapy. This is a detailed conversation facilitated by providing patient handouts, calendars, and specific printed instructions individualized to the patient. Incorporating medication administration into part of patients’ daily routine, using pillboxes, setting phone alarms or reminders, creating a schedule, and developing a tracking system are all strategies that can be tailored to individual patient needs [24,26]. Given the frequently changing therapy schedule, pharmacists can develop medication calendars for patients and schedule check-ins to assess patient understanding. Employing the “teach back” strategy, where the patient repeats the highlights of what they learned, is imperative to ensure the patient is comprehending these unique instructions. Pharmacists should also document, and revise patients’ medication lists to ensure it remains updated and accurate as patients’ transition throughout different cycles of therapy and rely heavily on updated available medication lists. AML patients are frequently admitted for complications, and regular pharmacist review and revision of the medication list is an essential step in preventing medications from being inappropriately continued or omitted in the inpatient setting. Furthermore, to prevent misunderstanding and errors at home, it is imperative that patients are given an updated medication list with associated written instructions with each visit. With frequent dose changes and starting/stopping preventative antimicrobials, patients are at high risk of errors if this level of detail is overlooked during patient encounters.

### 4.2. Empowering Patients to Advocate for Themselves

Since the administration of oral chemotherapy is not dependent on a patient’s physical presence in the clinic or infusion center, it becomes even more important for patients to self-identify common side effects. Furthermore, patients must understand how and when side effect management at home is appropriate versus what must be brought to their medical team’s attention. Although venetoclax is relatively well tolerated, pharmacists can recognize common hematologic and non-hematologic side effects and recommend prescription medications, over the counter (OTC) medications, and lifestyle modifications that can alleviate these symptoms, as well as, educate patients on how to monitor for and manage these side effects. Common side effects of venetoclax in combination with azacitidine therapy are listed in Table 2 [6,19]. Pharmacists can advise patients of important administration instructions, such as taking venetoclax regularly with their evening meal to facilitate drug absorption and perhaps minimizing gastrointestinal toxicities [19]. As previously addressed, hematologic side effects can be frequent, concerning, and challenging. Patients will likely develop neutropenia, anemia, and thrombocytopenia with severity that often warrants transfusion dependence [6]. Careful monitoring by all team members and patients themselves, in addition to patient adherence to appropriately scheduled laboratory checks, are imperative. The pharmacist can use shared decision making to increase patient activation by communicating the potential frequency of laboratory monitoring and transfusion support to patients when describing AML treatment options and helping patients make informed decisions that are properly influenced by their preferences and goals. Pharmacists can establish expectations for patients about the necessity of clinic visits, as they can be deemed as redundant and inconvenient to patients if they do not understand the importance and purpose of these visits.

## 5. Conclusions

Overall, care of cancer patients on self-administered therapies can be highly complex. As seen in the illustrative example of venetoclax in AML, a patient population that is often elderly, physically tenuous, and medically complicated, the risks for errors and non-adherence to therapy is significant. Oncology pharmacists are critical in navigating the health system-related, provider-related, and patient-related barriers of venetoclax-based regimens used in AML. One limitation of this narrative review is that is assumes that pharmacy resources discussed are widely available across medical institutions. However, if resources are not currently in place, this work can be used to serve as a blueprint for position justifications.

By simplifying the medication access process, facilitating guideline recommendations, managing therapy toxicities, breaking down complexities for patients, and supporting patients to be their own self-advocates, oncology pharmacists intervene at crucial steps throughout the medical process. These complex and multi-level pharmacist interventions can increase patient engagement and overall treatment success. Future evaluations of pharmacist driven processes with drugs such as venetoclax can help demonstrate both clinical and financial implications that derive from impacting patient and clinician behavior.

## Figures and Tables

**Figure 1 pharmacy-09-00130-f001:**
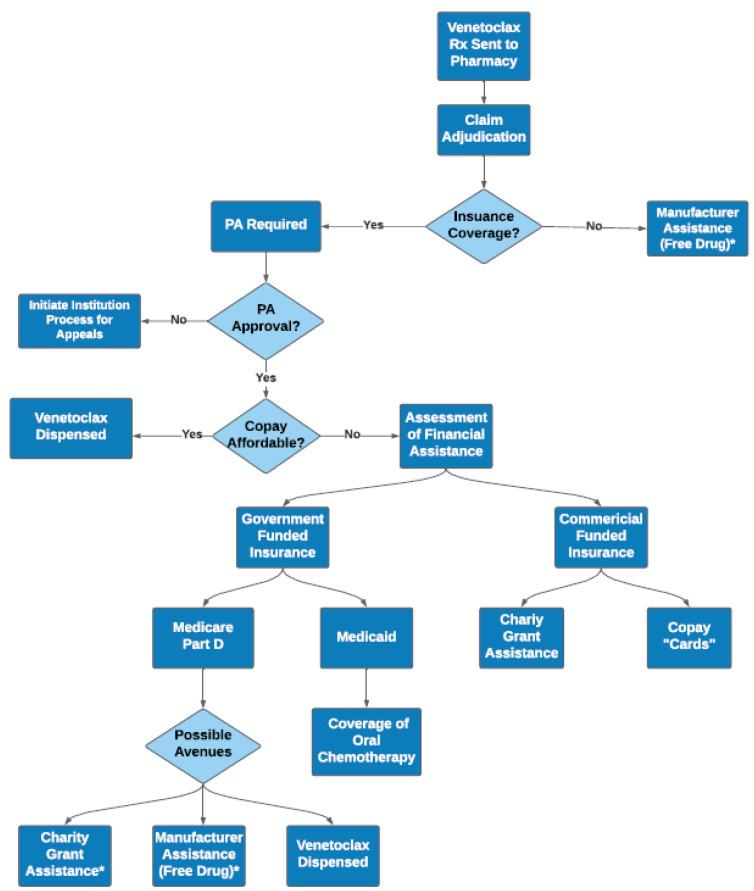
Complexities Surrounding Venetoclax Access in Clinical Practice. After venetoclax is sent to the pharmacy of choice, various pathways are available to help obtain medication and limit patient financial toxicity. Based on insurance status, copay affordability, income status, and other factors, patients may be eligible for copay cards, manufacturer assistance, charity grant, or other avenues for financial help. Rx = prescription; PA = prior authorization; * If patient meets income eligibility requirements for assistance.

**Table 1 pharmacy-09-00130-t001:** Major Drug-Drug Interactions [6,19,20].

Co-Administered Drug *	Effect on Venetoclax	Recommendation
**Strong CYP3A4 inhibitors** Posaconazole, voriconazole, itraconazole, indinavir, lopinavir/ritonavir, telaprevir, clarithromycin, conivaptan	Significant increase in concentration	↓ venetoclax dose by 75% **
**Moderate CYP3A4 inhibitors** Fluconazole, isavuconazole, erythromycin, ciprofloxacin, diltiazem, dronedarone, verapamil	Moderate increase in concentration	↓ venetoclax dose by 50%
**Strong CYP3A4 inducers** Carbamazepine, phenytoin, rifampin, St. John’s wort, avasimibe	Significant reduction in concentration	Avoid use
**Moderate CYP3A4 inducers** Bosentan, efavirenz, etravirine, nafcillin, modafinil	Moderate reduction in concentration	Avoid use
**P-gp inhibitors** Amiodarone, azithromycin, captopril, carvedilol, ticagrelor cyclosporine, quinidine	Moderate increase in concentration	↓ venetoclax dose by 50%
**P-gp substrates** Digoxin, everolimus, sirolimus	N/A	Avoid use; administer P-gp substrate at least 6 h before venetoclax
**Warfarin**	N/A	Monitor INR daily

N/A, not available; INR, international normalized ratio; ↓ decrease. * These lists are not exhaustive, and caution should be exercised when prescribing new medications in the setting of venetoclax. ** Venetoclax dose reduced to 70 mg in the setting of posaconazole specifically.

**Table 2 pharmacy-09-00130-t002:** Select adverse events associated with venetoclax and azacitidine therapy [6,19].

Adverse Event	All Grades	≥Grade 3
	Percentage of Patients	
**Gastrointestinal disorders**		
Nausea	44	2
Vomiting	30	2
Diarrhea	43	5
Constipation	43	1
Decreased appetite	25	4
Abdominal pain	18	<1
Stomatitis	18	1
**General disorders**		
Musculoskeletal pain	36	2
Fatigue	31	6
Edema	27	<1
**Skin rash**	25	1
Infections		
Sepsis	22	22
Urinary tract infections	16	6
Pneumonia	23	20
**Hemorrhage**	27	7
**Dyspnea**	18	4
**Dizziness**	17	<1
**Peripheral edema**	24	<1
**Pyrexia**	23	2
**Febrile neutropenia**	42	42
**Hypokalemia**	29	11
**Hematologic laboratory abnormalities**		
Neutropenia	42	42
Thrombocytopenia	46	45
Leukopenia	21	21
Anemia	28	26

## Data Availability

Not applicable.

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
