# Peer review of "Impact of Oncology Pharmacists on the Knowledge, Attitude, and Practices of Clinicians to Enhance Patient Engagement of Self-Administered Oral Oncolytics"

_pharmacy, 2021, doi:10.3390/pharmacy9030130_

Round 1

Reviewer 1 Report

The topic is interesting and worthy of investigation. It is about promoting patient activation and engagement in therapy in cancer care.

Even it is a review paper, it should be conceptualized as a scientific paper, taking more into account the IMRAD paper structure. We should know, if we repeat the review, how should we do it. 

Please, form the abstract in the following manner. First, describe the background of the research (1-2 sentences). Second, describe the goals of the research (1-2 sentences). Third, describe briefly (1-2 sentences) the methodology used. Fourth, describe the results and the conclusion of the research in 3-4 sentences.

In the last section, please focus on “Conclusion” to include
(1). Academic Implications
(2). Limitations of the paper
(3). Future Studies and Recommendations

Author Response

Thank you for your comments. Our responses are attached.

Reviewer 2 Report

This is an interesting review and generally it is well written. There are however a couple of areas which I believe need to be further addressed by authors for this review to reach its full potential.

From abstract, it is not clear if this is a review article or a case study. The title is therefore misleading potentially. Authors should make this clearer.  A review or a case study have different structure and pathways of presenting the paper. Therefore, I believe this is a major issue that authors should address accordingly.

In the introduction section, authors state “AML is the most common acute leukemia in adults with 19,940 new cases and 5311,180 deaths in 2020”. I would recommend they specify where do these figures apply.

I believe the introduction fails to fully introduce the theme that is discussed in this paper. I would recommend authors write a paragraph at the end of the introduction section to address this aspect.

Has there been any prior related work on this? Introduction section does not account for this very well. Authors can discuss similar contributions that pharmacists make with other clinical pharmacy services and then discuss how would that relate to the focus of this study in particular.

Towards the end of the introduction section, authors specify the 3 domains that pharmacists can impact. Authors then divide these 3 domains in subheadings at a same level as Introduction. I would suggest authors review this structure because it is a bit confusing.

Authors have used the term financial toxicity, which is fine, however I would suggest they define the term as it can be confusing for someone not strictly in this field.

In various sections authors have used ‘Shepherding Medication Access and Ensuring Rapid Initiation of Therapy’ or ‘Breaking Down Treatment Complexities’. I would suggest this is incorporated in the numbering system of the particular section to make it clearer.  

Whilst its clear that pharmacists can provide information on drug interactions to other health professionals and patients, I believe an important aspect to further expand is the inteprofessional and/or multidisciplinary work involved in the provision of venetoclax. I would suggest authors discuss this aspect as well.

Author Response

(The authors gave the same response as above.)

Reviewer 3 Report

Impact of an Oncology Clinical Pharmacist on Patient Engagement in Self-Administered Oral Oncolytic Therapy: Case Study of Venetoclax in Acute Myeloid Leukemia Patients

The report is well written but in my opinion to a large extent, went off on a tangent. By reading the abstract (shown below), I presume the review focuses on impact of an oncology pharmacist on patient engagement in self-administered venetoclax

Abstract

Oncology clinical pharmacists are uniquely positioned to make interventions to increase patient activation and engagement. To accomplish this goal, pharmacists can target health system-related, provider-related, and patient-related factors to help enhance patient-centered care and drive behavioral health changes. Interventions that pharmacists must tackle include educating team members and patients on the medication acquisition process, communicating urgency of treatment, optimizing workflows, facilitating guideline recommendations, preventing, and managing treatment toxicities, and promoting patient self-advocacy through education and shared decision-making. As crucial members of the healthcare team, oncology clinical pharmacists can simplify highly complex treatment regimens to facilitate and optimize patients’ ownership of their care. This review will focus on the example of venetoclax treatment in acute myeloid leukemia to demonstrate the impact that pharmacists provide that leads to behavioral change of patients and clinicians.

It was further stressed on in the Introduction that : However, for institutions without dedicated medication assistance teams, pharmacists are often critical  in facilitating proper communication between all stakeholders to ensure prompt access to  therapy. Pharmacists can help create a resource with a systematic approach that can help other team members understand the steps to medication access, to help with both patient education and more timely acquisition for the patient

However, after reading your manuscript, I was slightly disillusioned as the content focuses on:

  1. Health System-Related Domain

Shepherding Medication Access and Ensuring Rapid Initiation of Therapy

  1. Provider-Related Domain

Facilitating Guideline Recommendations

Navigating Therapy Toxicities

  1. Patient-Related Domain

Breaking Down Treatment Complexities

Empowering Patients to Advocate for Themselves

It seems that the author did not include reason why the drug Venetoclax was chosen ?

The title also does not seem to be appropriate: The case study vs review vs case series report ? This seems to be a narrative review

Other comments:

Suggesting the use of oncology pharmacist rather than oncology clinical pharmacist-as evident by argument showed below:

https://www.ncbi.nlm.nih.gov/pmc/articles/PMC2720370/

https://www.pharmacytimes.com/view/we-are-all-clinical-pharmacists

Pls check: Supplementary Materials: The following are available online at www.mdpi.com/xxx/s1, Figure S1: 378 title, Table S1: title, Video S1: title.

Author Response

(The authors gave the same response as above.)

Round 2

Reviewer 3 Report

All comments have been addressed by the authors. I would like to recommend acceptance of this manuscript.